# Intermethod Characterization of Commercially Available Extracellular Vesicles as Reference Materials

**DOI:** 10.3390/biom16010066

**Published:** 2025-12-31

**Authors:** Sumeet Poudel, Diane L. Nelson, James H. Yen, Yuefan Wang, Hui Zhang, Zhiyong He, Ashley Beasley Green, Wyatt N. Veerland, Thomas E. Cleveland, Sean E. Lehman, Kurt D. Benkstein, Bryant C. Nelson, Lili Wang

**Affiliations:** 1National Institute of Standards and Technology, Gaithersburg, MD 20899, USA; 2Department of Pathology, School of Medicine, Johns Hopkins University, Baltimore, MD 21231, USA

**Keywords:** extracellular vesicles (EVs), reference materials, EV characterization, particle size distribution (PSD), particle number concentration (PNC)

## Abstract

The National Institute of Standards and Technology (NIST) is developing analytical methods to characterize extracellular vesicles (EVs) to support the urgent need for standardized EV reference materials (RMs). This study used orthogonal techniques, cryogenic electron microscopy (Cryo-EM), particle tracking analysis (PTA), asymmetrical flow field-flow fractionation (AF^4^), and microfluidic resistive pulse sensing (MRPS), to evaluate particle size distributions (PSDs) and particle number concentrations (PNCs) of human mesenchymal stem cells (MSCs) and LNCaP prostate cancer cell EVs. Proteomic profiles were assessed by mass spectrometry (MS), and microRNA (miRNA) content of LNCaP EVs was evaluated by small RNA-seq at two independent laboratories. A commercial green fluorescent protein exosome served as a control, except in Cryo-EM, proteomic, and miRNA analyses. Cryo-EM, regarded as the gold standard for morphological resolution, served as PSD reference. PSDs from all methods skewed larger than Cryo-EM, with MRPS closest, AF^4^ most divergent, and PTA intermediate with broader distributions. All techniques reported broad PSDs (30 nm to >350 nm) with PNCs decreasing with increasing particle size, except for AF^4^. Quantitative discrepancies in PNCs reached up to two orders of magnitude across methods and cell sources. MS identified global and EV-specific proteins, including syntenin-1 and tetraspanins CD9, CD63, and CD81. RNA-seq revealed notable inter-laboratory variation. These findings highlight the variability across measurement platforms and emphasize the need for reproducible methods to support NIST’s mission of developing reliable EV reference materials.

## 1. Introduction

Extracellular vesicles (EVs) are a class of membrane-bound endogenous vesicles that shuttle complex cargo of proteins, lipids, and nucleic acids that alter function in both local and systemic recipient cells. EVs are heterogeneous in size, ranging between 40 nm and 5000 nm in diameter [1,2,3,4]. They are released from normal and pathogenic living cells and are detected in all varieties of human body fluids. Based on the biogenesis classification, two main subtypes of EVs are ectosomes (plasma membrane-derived) and exosomes (endosomal compartment-derived) [5], which are differentiated based on their biogenesis, release pathways, size, content, and function. So far, extensive evidence on all these types of vesicles indicates that EVs play a key role in the intercellular communication between cells, along with the secretion of small soluble molecules and direct cell–cell contact [3,6]. For this reason, EVs and their associated cargo are considered promising candidates for biomarkers in clinical applications [7,8]. Additionally, EVs can be utilized as drug-delivery vehicles by being modified to contain specific therapeutics. The lipid bilayer membrane of the EVs limits the degradation of the cargo and is highly selective in cellular targeting [9]. To successfully use EVs as therapeutics, a scalable source of well-characterized EVs, proper classification, and thorough characterization are key metrics. The classification of EVs in various subtypes has been based on size, biogenesis, separation methods, origin, or function [10,11,12,13,14,15]. However, the scalable and reproducible characterization and classification of EV subtypes will require valid single-particle characterization techniques capable of phenotyping surface molecules and molecular signatures [16].

Single-particle methods are necessary to deconvolute EV heterogeneity, enabling the robust and reproducible determination of population size distributions and concentrations. Nonetheless, the detection and evaluation of single EVs are challenging because of their heterogeneity in size and cargo composition, low refractive indices (<1.5) [17], and their co-existence with other biological molecules that are non-EV in nature, like protein aggregates, cell debris, or viruses [18,19,20]. The determination of concentrations also relies on factors such as the accuracy of volume intake for concentration measurement and the detection limit of sizes by the instrument [21]. To this end, EV measurements must be standardized, and reference materials (RMs) must be developed [1,21,22]. The first annual meeting of ISEV identified the need to establish standards or RMs for clinical certification and manufacturing of EV-based diagnostic kits and therapies [23].

NIST defines an RM as a material or substance whose one or multiple property value(s) are sufficiently homogenous and well established to calibrate an apparatus, assess a measurement method, or assign values to materials [24]. NIST has been actively investigating techniques to measure the integrity and properties of EVs. This work addresses some challenges that limit the broad use of EVs for clinical applications. Among the limiting factors are the lack of standardized cell-based platforms to produce EV-based therapeutics, the lack of EV RMs that allow researchers/manufacturers to validate EV measurements, and the lack of standardized measurement procedures for determining EVs’ size and molecular composition [21].

In this study, we characterized two candidate human-cell-derived EV RMs produced by the American Type Culture Collection (ATCC) and a fluorescent exosome control material based on their physical attributes (morphology, size, and concentration) and their cargo (EV proteins and small RNAs). To build capabilities of EV measurements and to assess whether commercial EVs can be used as an RM, two cell line-derived EVs (one from cancer cells and one from stem cells) and a green fluorescent protein exosome preparation (Millipore Sigma [25]) as a control material were obtained. The GFP labeling in the exosome preparation is generated through a viral gag-eGFP fusion, meaning the material behaves like viral-like particles (VLP) rather than physiological EVs because gag is not a natural EV cargo. For this reason, we used the material solely as a technical reference to assess detection sensitivity, assay linearity, instrument comparability, and nanoparticle assay benchmarking, and henceforth referred to as GFP VLPs. Thus, GFP VLPs are used to support assay performance, rather than to draw biological conclusions. All materials were sourced based on their market availability during the study. The sample EVs used in the study were hTERT MSC (hTERT-immortalized mesenchymal stem cells) and LNCaP (prostate carcinoma). The technology platforms for the study of particle size distributions (PSDs) and particle number concentrations (PNCs) were cryogenic electron microscopy (Cryo-EM), particle tracking analysis (PTA), asymmetric flow field-flow fractionation (AF^4^), and microfluidic resistive pulse sensing (MRPS). EV morphology was evaluated using Cryo-EM. EV-relevant proteins were identified using data-independent acquisition mass spectrometry (DIA-MS) for both cell lines. EV-specific small RNA (microRNA (miRNA)) populations were identified utilizing small RNA sequencing (RNAseq) platforms at two independent facilities with only LNCaP prostate cancer EVs as a candidate EV sample.

## 2. Materials and Methods

### 2.1. Extracellular Vesicles

Two candidate cell-derived EV RMs were acquired from ATCC (Manassas, VA, USA): hTERT MSC (ATCC #SCRC-4000-EXM) and LNCaP (ATCC #CRL-1740-EXM) and were provided as a frozen aliquot in dry ice. They were stored at −80 °C until use. According to the manufacturer’s product sheet, these EVs were isolated from their respective cell culture medium, purified, and concentrated using a tangential flow filtration (TFF) system in a sterile environment. Green fluorescent protein exosome (SAE0193, Lot #000023251) was acquired from Millipore Sigma (Burlington, MA, USA) in powder form, lyophilized from PBS, pH 7.4. The material was resuspended in 100 µL ice-cold ultrapure water. Similarly, according to the manufacturer’s product information sheet, GFP VLPs were extracted from HEK293T cells, which express GFP on the surface of their membrane. All methods, except for the Cryo-EM measurements, were replicate (duplicate or triplicate) measurements; any deviations are specified in the specific sections below. Samples from manufacturer vials were aliquoted into smaller volumes depending on acquisition requirements, and three independent aliquots were run on the same day for data acquisition. The sample volume for acquisition was based on manufacturer recommendations.

### 2.2. Microfluidic Resistive Pulse Sensing

MRPS measurements were collected using an nCS1 instrument (Spectradyne, Torrance, CA, USA; hardware version: 2). The sample holder and microfluidic chip were contained in disposable, polydimethylsiloxane cartridges. Specifically, the C-400 (65 nm to 400 nm diameter size range) cartridges were used for EV measurements. The limit of detection (LoD) is cartridge dependent and was set to 65 nm (measurement cutoff). Each lot of cartridges was pre-calibrated by Spectradyne for size and number concentration. A running buffer of phosphate-buffered saline (PBS) with 1% (*v*/*v*) polysorbate 20 (Tween 20, Sigma-Aldrich, St. Louis, MO, USA) was filtered (Thermo Scientific, Waltham MA, USA; Nalgene #565-0020; 200 nm) and used in the post-nanoconstriction flow channel to enable appropriate flow and washing. The measured samples were made by diluting the stock suspensions with PBS and the appropriate amount of polysorbate 20, such that the samples contained 1% (*v*/*v*) polysorbate 20 at number concentrations between 5 × 10^8^ particles/mL and 3 × 10^9^ particles/mL. EVs stored in 1% Tween 20 at 4 °C for up to 48 h exhibited minimal degradation, effectively retaining their size and structure [26,27,28]. The diluent was filtered through a 200 nm pore filter, followed by a 20 nm pore filter (Cytiva, Marlborough, MA, USA #6809-2002) before use. Collected data were analyzed using nCS1 Data Analyzer Software (version: 2.5.0.325). The peak filters used in this work are—Transit time (µs) < 100, Symmetry > 0.2 but <4.0, Diameter (nm) > 65 nm, Signal to noise ratio (S/N) > 10. The peak filters were used per the manufacturer’s recommendation.

### 2.3. Particle Tracking Analysis

PTA measurements were collected using a ZetaView 110 (Particle Metrix, Ammersee, Germany). The instrument was equipped with a laser (λ = 405 nm) to illuminate the sample. Instrument performance (size) was checked daily with a 110 nm polystyrene bead sample in scatter mode. Videos were recorded at 11 positions within the sample cell using a CMOS (Complementary Metal Oxide Semiconductor) camera (640 pixels × 480 pixels) through a 10×/0.30 NA objective. For the analysis of the initial set of EV samples (ATCC, Manassas, VA, USA), the camera shutter speed was set to 50 (corresponding to a 1/50 s exposure time), sensitivity to 80, and frame rate to 30 Hz. Videos were analyzed using the ZetaView software (version 8.04.02 SP2). In general, videos from all 11 positions were used in the analysis, but in several cases, certain positions were excluded from the study owing to high background scatter. At least 8 of 11 positions were used for cases where positions were excluded. Videos at each position consisted of 90 frames. The minimum track length was set to 45 frames. The minimum brightness (threshold) was set to 20 (maximum brightness 255), particle minimum area to 8 pixels, maximum area to 3000 pixels, and maximum track radius (jump distance) to 5 pixels. Particles appearing after the first frame were not tracked. Subsequent measurement of the GFP VLP sample used updated acquisition parameters based upon those used to analyze a silica particle dispersion (80 nm nominal diameter, nanoComposix, San Diego, CA, USA): camera shutter speed of 70 (1/70 s exposure time), camera sensitivity of 90, and camera frame rate of 30 Hz yielded size and number concentration values consistent with the manufacturer data sheet, allowing for the presence of doublet silica particles in the mixture. The daily (size) performance check employed a 122 nm polystyrene bead sample. Analysis settings were similar to what was used for the ATCC samples, with the minimum and maximum pixel areas adjusted to 9 px and 9999 px, respectively. Other analysis settings were unchanged. For these experiments, the measurement cutoff is closer to the probability of detection and depends on the sample and instrument setup. Here, the reported measurement cutoff (25 nm) is the lowest diameter detected and tracked.

EV samples were diluted gravimetrically with PBS, pH = 7.4. Reported EV size statistics from PTA, mean diameter, and standard deviation, were based on the sub-population of particles included in the PSD analysis. The reported PNCs for the sample dilution have been corrected. Uncertainties for diameter and PNC are reported as one standard deviation for *n* = 3 measurements.

### 2.4. Particle Asymmetric Flow Field Flow Fractionation-Multi-Angle Light Scattering

An Eclipse DualTec AF^4^ separation system (Wyatt Technology, Santa Barbara, CA, USA) was interfaced with an Agilent HPLC system (Model 1260, Agilent Technologies, Santa Clara, CA, USA) including a UV/Vis diode array detector (Model 1260, Agilent Technologies, Santa Clara, CA, USA), a HELEOS-II multiangle light scattering instrument (HELEOS-II, Wyatt Technology, Santa Barbara, CA, USA), and a differential refractive index detector (Optilab T-Rex, Wyatt Technology, Santa Barbara, CA, USA). The separation channel was a vendor-supplied “short” channel with a Mylar^®^ 250 mm thick “wide” spacer, and a 10 kDa nominal molecular-weight cut-off Ultracel^®^ regenerated cellulose (Millipore, Burlington, MA, USA) ultrafiltration membrane served as the accumulation wall. Samples were introduced into the AF^4^ separation channel via an autosampler (Model 1260, Agilent Technologies, Santa Clara, CA, USA) with a focus position of 12% of the channel length. The focusing was accomplished by flowing 0.2 mL/min of buffer into the channel inlet and 1.3 mL/min of buffer through the channel outlet for 5 min. After the samples were introduced and focused against the ultrafiltration membrane, they were eluted from the column in a size-selective manner with a 1.0 mL/min channel flow. In comparison, the cross flow was linearly ramped down from 1.0 mL/min to 0 mL/min over 60 min. For post-separation, the channel was rinsed for 5 min with 1.0 mL/min channel flow and 0 mL/min crossflow with the injector “on” to rinse out the sample loop. The fluid medium for the separation was an isocratic phosphate-buffered saline solution (Sigma Aldrich, St. Louis, MO, USA, P3813) with 3.07 mmol/L sodium azide (Ricca Chemical, Arlington, TX, USA) as a preservative. The eluted fractions flowed into the HELEOS-II detector, where the flow cell was illuminated with a plane-polarized laser (λ = 662 nm), and the scattering intensity was measured at 16 different angles simultaneously. Bovine serum albumin (BSA, Thermo Fisher Scientific, Waltham, MA, USA) was used as a system check when setting up a new membrane. After the membrane was set up and passivated, an analytical BSA injection was used as a quantitative check. The recovery of the BSA was used to assess membrane passivation, and the MALS-calculated molar mass was used to ensure the system was working correctly. Data analysis and calculations were done using ASTRA 7.3.2 software. The scattering intensities, scattering angles, and laser wavelength were used to determine the diameters of the monodispersed spherical particles. The EV data was modeled using a coated-sphere model consisting of a sphere with a refractive index of 1.3540 with a 5.5 nm shell with a refractive index of 1.4000 [29]. The volume and refractive indices of the particle and the medium were used to determine the particle number count in each measurement sample volume. The measurement cutoff for AF^4^ for these experiments was set to 30 nm diameter.

### 2.5. Cryo-Electron Microscopy

EV suspensions were thawed and kept at 4 °C or on ice until application to grids for electron microscopy. Grids used for quantifying EV diameters were ultra-thin continuous carbon with lacey carbon supports (Ted Pella #01824G), as well as R1.2/1.3 Cu200 and R3.5/1 Cu200 (Quantifoil, Großlöbichau, Germany). Grids were glow-discharged using a Pelco easiGlow system at 25 mA for 25 s with a pre-process hold time of 10 s. Grids were suspended in the chamber of a Vitrobot Mark IV system (Thermo Fisher Scientific, Waltham, MA, USA) set to 4 °C and 100% humidity, and then 3 µL of sample was applied before blotting and plunge-freezing into liquid ethane. Grids were then transferred to a Talos Arctica microscope (Thermo Fisher Scientific) and imaged at 200 kV with a Falcon 3EC direct electron detector using the software EPU.

For EV diameter quantification, dose-fractionated movies were collected at a nominal magnification of 11,000×, corresponding to a calibrated pixel size of 0.96 nm +/− 0.01 nm. Pixel size calibrations were checked periodically against a cross-grating replica (Ted Pella #673) with an accuracy of 1%. Movies were collected in linear mode, using 133 frames, a total exposure time of 20 s, and a total dose of 2000 e/nm^2^. An objective aperture of 30 µm was used for data collection, with target defocus values of −5 µm to −15 µm. Motion correction was performed using MotionCor2 [30]. Images were analyzed with Fiji/ImageJ (software version 2.14.0/1.54) by manually outlining each EV via freehand selection, quantifying the area enclosed, and converting this to a diameter (assuming a sphere). For each cell line, approximately 500 single EVs (without internal structures) were counted across multiple images, and counting stopped once this number was reached. Selection criteria required EVs to have a diameter greater than 30 nm and no overlap with thicker carbon support structures or other EVs. The 30 nm diameter cut-off was chosen because smaller EVs could not be reliably distinguished from grid contamination and artifacts.

### 2.6. Particle Size Distribution Statistics

All methods, except for the Cryo-EM measurements, were performed in triplicate. EV diameter measurements were grouped into 5 nm bins, ranging from 0 nm to 400 nm. The bin size was chosen to accommodate one of the methods and applied to the other for direct comparison. The statistical analysis treats the diameters as multinomial data using the bins.

For each diameter bin, the probability for *bin* i is $p_{i}=\;\frac{x_{i}}{N}$, where $x_{i}$ is the count in that bin, and *N* is the total number of diameter measurements in that replicate. The estimated $p_{i}\;$values are dependent on the binning scheme. The uncertainty of $p_{i}\;$can be estimated from the theoretical multinomial formula [31] for the standard deviation of $p_{i}\;$, which is $\sqrt{\;p_{i}\;(1-p_{i}\;)/N}$. The relative uncertainty (referred to as single measurement relative uncertainty) is calculated by dividing this uncertainty by $p_{i}$.

For each bin, the mean $p_{i}\;$is the average of the three probabilities for each replicate for that bin and has an associated relative uncertainty, which is calculated using a bootstrap analysis [32]. This analysis simulates sampling 100,000 times from three multinomial distributions with the same bin probabilities as the actual replicates and then taking the standard deviation of the simulated sample of mean bin probabilities for each bin. For each bin, this relative uncertainty is largely dependent on the replicate-to-replicate variability of the bin probabilities for that bin. Thus, a large uncertainty for a mean $p_{i}\;$shows that there is disagreement between the replicates for the estimated $p_{i}\;$for that bin. For this reason, this mean relative uncertainty will be referred to as replicate-to-replicate relative uncertainty.

An example of the computer code has been provided in the Appendix A section.

### 2.7. Particle Number Concentration Measurements

EV PNCs were measured in triplicate and reported as mean ± one standard deviation.

### 2.8. ESI-LC-MS/MS for Global Proteome Data-Independent Acquisition (DIA) Analysis

Aliquots of both EV isolates (*n* = 3; 13 µg total protein/sample) were digested in solution [33], and the resulting peptides were analyzed using an Orbitrap Exploris 480 mass spectrometer. Total protein concentrations in the EV isolates were determined using Qubit measurements (Thermo Fisher Scientific, Waltham, MA, USA) according to the manufacturer’s protocol. Digested peptide material from individual samples was spiked with index Retention Time (iRT) peptides (Biognosys, Schlieren, Switzerland, 0.5×) and subjected to DIA analysis. Approximately one µg of peptides was separated on an in-house packed 28 cm × 75 mm diameter C18 column (1.9 mm Reprosil-Pur C18-AQ beads (Dr. Maisch GmbH, Ammerbuch, Germany); Picofrit 10 mm opening (New Objective)), lined up with an Easy nLC 1200 UHPLC system (Thermo Scientific, Waltham, MA, USA). The column was heated to 50 °C using a column heater (Phoenix-ST, Chadds Ford, PA, USA). The flow rate was set at 200 nL/min. Buffers A and B were 3% acetonitrile (0.1% formic acid) and 90% acetonitrile (0.1% formic acid), respectively. The peptides were separated with a 7% to 30% B gradient in 85 min. Peptides were eluted from the column and nano-sprayed directly into Orbitrap Exploris 480 mass spectrometer (Thermo Scientific, Waltham, MA, USA). The mass spectrometer was operated in a DIA mode. The DIA segment consisted of one MS1 scan (400 *m*/*z* to 1000 *m*/*z* range, 60 K resolution) followed by 20 MS2 scans (fixed *m*/*z* range, 30 K resolution). Additional parameters were: MS1: RF Lens at 34%, AGC Target 1.0 × 10^6^, charge state include 2 to 6; MS2: window Overlap (*m*/*z*) at 0.7, Max IT at 25 ms to 60 ms [34].

### 2.9. Spectral Library Generation for DIA Analysis

For the proteomics spectral library generation, each sample had two replicates of data-dependent acquisition (DDA) data and three replicates of DIA data. The library consisted of twenty-five raw data files. Raw mass spectrometry files from DDA and DIA platforms were processed using Spectronaut (Biognosys, Schlieren, Switzerland) to generate a combined spectral library that integrated DDA and DIA search results. The DIA data was searched using the default setting of Spectronaut with precursor and protein Q-value cut-off set at 1%. The results were exported with normalization as Cao et al. [34] detailed. The normalized MS peak intensities were calculated using Equations (1) and (2) below, where K  is the mean data for each raw DIA sample file/protein (n = 3), *x* is the raw data measurement, $\bar{a}$ is the mean of the raw DIA data/protein, $\bar{C}$ is the mean of the protein data or the raw DIA sample file, $\bar{D}$ is the mean of all data measurements and $\bar{K}$ is the final normalized mean data for each raw DIA sample file/protein (n = 3). A table of EV-associated marker proteins with their relative expression profiles for the 2 EV samples can be found in the Appendix A.(1)Normalized Data K=xa¯C¯D¯(2)$$Final\;Normalized\;Data=\left(\bar{K}\right)$$

### 2.10. Small RNA Extraction and Sequencing

Using LNCaP EVs as a candidate for small RNA analysis, total small RNA was extracted using Qiagen’s miRNeasy Tissue/Cells Advanced Mini Kit (Venlo, The Netherlands, Cat# 217684) via two independent RNA sample extractions performed by two individuals. After RNA was eluted from the column with 50 µL elution buffer, the RNA samples were transferred into RNase-free tubes in 10 µL/tube aliquots. The RNA concentrations were assessed by nanodrop using the A280/A260 and A260/A230 ratios. The total RNA sample was quantified using an Agilent Bioanalyzer 2100 instrument (Santa Clara, CA, USA) with a corresponding Agilent Bioanalyzer Small RNA Analysis kit (Part # 5067-1548) for the analysis and quantitation of small RNA samples containing (6 to 150) nucleotides (nts). The resulting RIN (RNA integrity number) scores and concentrations were considered when qualifying samples to proceed. Two aliquots from vial #1 were named Sample #1 and Sample #2, and 2 aliquots from vial #2 were named Sample #3 and Sample #4, and shipped to two independent facilities (Facility A and Facility B) for sequencing. The four samples (1, 2, 3, and 4) are identical. cDNA library preparation and sequencing were performed at the two facilities. An Illumina MiSeq (San Diego, CA, USA) platform was utilized at Facility A with 30 million single-ended reads per sample. An Illumina NovaSeq 6000 (San Diego, CA, USA) was used at Facility B with 20 million single-ended raw reads per sample. The resulting fastq files from both facilities were used in subsequent analysis at the bioinformatics core of Harvard T.H. Chan School of Public Health.

### 2.11. Small RNA Sequencing Fastq Analysis and Reporting

LNCaP EV small RNA sequencing from 2 facilities generated fastq files. The resulting fastq files from both facilities were used in subsequent analysis. A Appendix A (https://rpubs.com/snp34/1293230, accessed on 9 December 2025) has also been created by using basic R [35] and its statistical tools which houses: raw tables that list small RNA (miRNA and genic) for LNCaP (prostate carcinoma) EV sequencing from 2 facilities (facilities A and B), basic comparison of data from facilities A to B along with a comparison to EVAtlas database [36] for miRNAs. The packages and codes for the R analysis are in the report.

## 3. Results

### 3.1. EV Morphology

Cryo-EM allows for direct, high-resolution visualization of EVs in their native, hydrated state [37]. Figure 1 shows characteristic Cryo-EM images of the MSC (Figure 1a) and LNCaP (Figure 1b–e) EVs. Cryo-EM analyses confirmed the presence of EVs within each sample, showing EVs of various sizes and morphologies. Multilayer EVs (Figure 1d) and EVs within other EVs (Figure 1c) are distinguishable from single EVs (Figure 1b) and asymmetrical contaminants (Figure 1e, red arrows). EVs exhibit a prominent “rim” corresponding to the bilayers, whereas contaminating objects do not possess this feature. GFP VLPs have been widely used as a biological RM in the EV field [25]. These GFP VLPs function as viral-like technical reference particles rather than physiological EVs, so we use them only for evaluating assay performance, and we recommend interpreting their results accordingly. A single Cryo-EM measurement was performed on the other two EV sample preparations.

### 3.2. Particle Size Distribution by Method

Due to the low sample throughput of current Cryo-EM platforms, other analytical techniques should be considered for determining PSDs of EV samples. Cryo-EM is the primary technique for measuring morphological size for EVs, albeit in a low-throughput setup, while other analytical techniques measure hydrodynamic sizes. Hydrodynamic sizes may be larger than morphological sizes due to the presence of a protein corona, hydrated layers, and other surface-associated molecules [38]. GFP VLP was used as a control material to compare the PSDs across orthogonal measurement techniques: MRPS, PTA, and AF^4^ [39]. As GFP VLPs [25] has been routinely used as an exosome RM for EV, it was implemented as an external control to better compare the different measurement techniques after the initial analysis of hTERT MSC and LNCaP EVs. Figure 2a-left shows the mean PSD for GFP VLPs measured in triplicate. The mean PSD for MRPS shows a dominant population in the smaller range, followed by a systematic decrease in bin probability for increasing particle size. MRPS has an instrument cut-off at 65 nm. The mean trendline follows an asymmetric distribution skewed towards larger particles with a mean diameter of 169 nm (81 nm standard deviation; Appendix A). The mean PSD for PTA shows a broad distribution with a mode at approximately 105 nm with a long tail extending for larger particles, while AF^4^ asymmetric distribution skewed towards smaller particles and shifted furthest to the larger particle sizes as compared to other methods (Figure 2a). The mean (standard deviations) diameters were 115 nm (36 nm) and 193 nm (11 nm) for PTA and AF^4^, respectively. Figure 2a-center shows the single measurement relative uncertainties for each independent replicate for each method. The single measurement relative uncertainty shown is the estimated uncertainty for each bin probability divided by that bin probability. Because of this division by the bin probability, an inverse correlation can be expected between the mean bin probability and relative uncertainty. For GFP VLPs, this relationship holds for PTA and MRPS. The single measurement relative uncertainty for AF^4^ was near zero because of the large number of particles measured in AF^4^ (1 × 10^9^ to 1 × 10^10^) compared to the other techniques (1 × 10^3^ for MRPS and PTA) (see Appendix A). Figure 2a-right depicts the replicate-to-replicate relative uncertainties for the mean distributions of GFP VLPs. For MRPS and PTA, the replicate-to-replicate relative uncertainty overlapped and followed the expected inverse correlation between mean bin probability and relative uncertainty. The replicate-to-replicate relative uncertainty (Figure 2a-right) was higher for AF^4^ in its range of measurement due to the dissimilarity of the single measurements (Appendix A).

For LNCaP and MSC (Figure 2b,c, left), the mean PSDs measured by Cryo-EM were considered the most accurate and reliable. However, specimen preparation, blotting, and vitrification might influence the EV number and size distribution [40]. The PSD obtained from MRPS closely resembled that of Cryo-EM, with a shift towards larger particles, a larger population in the smaller range, and a gradual decrease in bin probability for increasing particle size. For LNCaP, the mean (standard deviations) diameters were 54 nm (17 nm) and 86 nm (22 nm) for Cryo-EM and MRPS, respectively. For MSC, the mean (standard deviations) diameters were 61 nm (19 nm) and 79 nm (22 nm) for Cryo-EM and MRPS, respectively. Like GFP VLPs, the mean PSD measured by PTA for LNCaP and MSC showed broad distribution with long tails for larger particles, and their mean (standard deviations) diameters were 130 nm (32 nm) and 115 nm (36 nm), respectively. With LNCaP, the AF^4^ measurement yielded a mean PSD with an asymmetric distribution with a shorter rightward skew compared to Cryo-EM and MRPS; mean (standard deviations) diameter was 182 nm (11 nm). While MSC AF^4^ measurement yielded three separate peaks, mean (standard deviations) diameters were 144 nm (7 nm), 184 nm (5 nm), and 162 nm (10 nm). For LNCaP and MSC, as expected, the single measurement relative uncertainty (Figure 2b,c, center) was inversely correlated to the mean bin probability for each method except AF^4^, which was near zero because of the large number of particles measured (Appendix A). The replicate-to-replicate relative uncertainty overlapped for Cryo-EM, MRPS, and PTA (Figure 2b,c, right) and followed the expected inverse correlation between mean bin probability and relative uncertainty. For LNCaP, the replicate-to-replicate relative uncertainty (Figure 2b-right) was lower for AF^4^ due to the similarity of the single measurements (Appendix A). The uncertainty for MSC (Figure 2c-right, and Appendix A) was higher compared to other measurements. The replicate-to-replicate relative uncertainty for AF^4^ differed for all three cell lines (Figure 2a–c, right).

For MRPS, an incorrect calibration of the orifice diameter by the manufacturer or a diluent with different conductivity could both explain the systematic shift to the right compared to Cryo-EM. PTA has a larger size distribution than MRPS, particularly for polydisperse samples, because PTA relies on the optical detection of light scattered from multiple particles simultaneously. In contrast, MRPS measures changes in electrical resistance as particles pass through a pore. Light scattering intensity, which is sensitive to the particle’s size and refractive index (RI), scales with radius to the 6th power (in the Mie regime, rough approximation), while the electrical resistance signal (dependent on particle volume) scales with radius to the 3rd power [34,35]. Thus, PTA can more readily detect and track larger particles in a polydisperse sample due to the scaling of scattered intensity for a given instrument setting [41]. Uncertainties associated with PTA number concentrations and size distributions also include the probe volume, the size LoD, uncertainty in the size LoD with biases against larger particles, and a lack of specificity, because non-vesicle particles also scatter light (note, though, fluorescent-PTA approaches with labeled EVs are an active area of research [42,43]). The lower probabilities for diameters smaller than the peak diameter (also note the rapidly increasing uncertainties associated with bins smaller than the peak diameter, Figure 2) are likely due to reduced detection, counting, and tracking of smaller particles. While adjusting sample conditions and PTA settings (e.g., decreasing shutter speed, increasing camera sensitivity, and/or increasing frame rate) may improve coverage, optimizing these settings remains challenging without reference or calibration materials specific to EV RI and size. A sample containing a mixture of low RI EVs of various sizes is likely to exhibit larger variability between measurement methods [41].

In contrast, AF^4^ (Figure 2a–c, left and right) did not produce overlapping PSD across the replicates, except for LNCaP. Although single measurement uncertainty was low (Figure 2a–c, center), the differences between the replicates (Appendix A) led to minimal overlap in the measurements (Figure 2a,c, left) and a large replicate-to-replicate uncertainty, except for LNCaP EVs. The PSD calculated from MALS measurements after AF^4^ separation has been shown to differ significantly from other size measurement techniques previously, albeit for polystyrene nanoparticles [44]. MALS measurements require accurate knowledge of both particle volume (shape and size) and particle refractive index (particle count), thus any inaccuracy in particle size measurement from MALS calculations or the low scattering intensity due to the low refractive index of EVs will substantially increase the inaccuracy of the measurement [17,29,41]. Furthermore, the AF^4^ measurement process significantly dilutes the sample, thus requiring very high concentrations (1 × 10^10^/mL to 1 × 10^12^/mL) of each particle diameter for accurate counting in the flow cell. With PNCs of approximately 1 × 10^6^/mL distributed across the size range, the average number of particles inside the flow cell during measurement is about one. This leads to higher variations in the data based on a low signal-to-noise ratio. It is worth noting that AF^4^, as a technique, required larger sample volumes (about 1.5 mL per triplicate run) compared to the other methods (<50 µL at 1 × 10^7^/mL to 1 × 10^9^/mL).

Cryo-EM measurements for the LnCaP and MSC EVs capture the smallest particles; the shape of PSD from MRPS closely resembles the PSD of Cryo-EM, although shifted to larger particle sizes, while AF^4^ PSD were shifted still further to the largest sizes. The shape of the PTA PSDs showed differences from Cryo-EM and MRPS PSDs, with a broad distribution profile centered at >100 nm, and a long tail extending to larger particle sizes. These PTA measurements were made in a different, larger size range; furthermore, PTA more readily detects and tracks the larger particles that scatter more light [41,45]. The AF^4^ mean PSD analysis yielded three variable profiles for each EV analyzed, which was the least reproducible method.

### 3.3. Particle Size Distribution per Cell Line-Derived EV

The mean PSD from each cell line was compared using a single method to gain deeper insight into the variability of samples across instruments. Overall, Cryo-EM, MRPS, and PTA (Figure 3a–c) have low between-sample variability when assessing the precision of repeated consecutive measurements on a group of particles under identical conditions [46]. Cryo-EM (Figure 3a) and MRPS (Figure 3b) display a similar asymmetric and skewed to the right trendline, with Cryo-EM measured particles ranging from 30 to 185 nm and MRPS measured particles ranging from 65 nm to 350 nm. For PTA (Figure 3c), all cell lines produced a broad distribution with a tail towards larger particles and particles ranging from 25 nm to 390 nm. We noticed 2 to 3 local maxima in the broad distribution of the LNCaP sample, and we attribute this to the binning strategy. Figure 3d shows that AF^4^ has high between-sample variability for the MSC EVs. Both the GFP VLPs and the MSC EVs had different peaks for each replicate (Appendix A), although for the GFP VLPs, the three peaks are close enough to make the average resemble a single peak. For MSC EVs, two peaks are close enough to make the average look bimodal. For the LNCaP EVs, the replicate distributions are closer but follow an asymmetric distribution that is skewed to the left. The distribution of EVs measured by AF^4^ were within 90 nm and 250 nm with multiple peaks for GFP VLPs and MSC EVs, signifying AF^4^ may be detecting EV subpopulations.

### 3.4. Particle Number Concentration per Cell Line-Derived EV per Method

Table 1 lists the PNCs (1/mL) of the control GFP VLPs, and each cell line-derived EV as measured by PTA, AF^4^, and MRPS. Cryo-EM was not used to measure EV PNC due to specimen preparation conditions potentially causing aggregation and preferential binding of EV to the support structure [40]. The measurement cutoffs were included for each instrument because EV PNC depends on the lower LoD [47]. The GFP VLPs PNC was listed as 3 × 10^9^/mL for Lot #0000232513 on the manufacturer’s certificate of analysis (CoA). Note: the manufacturer’s CoA does not provide the uncertainty on the PNC. PTA and MRPS measured PNCs in the same order of magnitude while AF^4^ yielded one order higher PNC for GFP VLPs. However, AF^4^ PNCs for LNCaP and MSC EVs were one to two log orders lower than those measured by PTA and MRPS. Specifically, estimates of PNCs for LNCaP and MSC were on the order of 10^12^ for PTA and varied by one order of magnitude for AF^4^ (10^10^ and 10^11^, respectively) and MRPS (10^11^ and 10^12^, respectively). MALS was utilized for the measurement of the AF^4^ PNC, and it relies strongly on the difference in RI between the particle and the medium. The low RI of EVs makes the determination of the PNC less accurate [17,29]. Furthermore, EVs could bind to the membrane used in AF^4^ which would also impact PNC measurements. Although MRPS is not sensitive to RI, unlike AF^4^ and PTA, the cartridges are prone to clogging which directly impacts the sampling volume in addition to a 65 nm particle size cut-off limiting the accuracy in EV PNC determination [41]. Nominal estimates of the PNCs for the LNCaP and MSC EV samples were not provided by the manufacturer; thus, the accuracy of the tested methods for EV PNC requires further investigation.

### 3.5. Proteomic Profiling of MSC and LNCaP-Derived EVs

Equivalent masses of the sample (based on 13 µg of total protein) were digested for the MS analyses. However, during the preanalytical phase of sample preparation, we found that the LNCaP EV sample was notably lower in protein levels (0.34 µg/µL) in comparison to the MSC EV sample (7.06 µg/µL). Regarding the number of EVs per µg protein for each sample, we found the following: LNCaP = 2.1 × 10^9^/µg protein; MSC = 1.4 × 10^8^/µg protein. EV particle numbers are based on the MRPS values reported in this manuscript. The number of EVs per µg protein for the MSC samples is lower than for the LNCaP samples. It is conceivable that the differences in protein levels between the samples are due to factors like protein aggregation, the presence of ribonucleoprotein complexes, or lipoproteins. DIA-MS analysis of the MSC and LNCaP EVs returned 733 and 1867 unique proteins, respectively. The difference in EV-associated markers and proteomics profiles between the two samples may be attributed to the differences in their EV biogenesis pathways, or the donor cell types (cancer cell for LNCaP vs. stem cell for MSC). The EV proteomic database [48] search identified approximately 131 EV-associated/marker proteins that were commonly expressed in 1 or both cell types (Appendix A). Not surprisingly, the protein expression profiles were highly variable between the LNCaP prostate cancer cell line and the MSC EVs, as illustrated in the four heatmap panels shown in Figure 4.

Some of the most frequently utilized EV marker proteins identified in eukaryotic cell studies are identified with a black star (Figure 4a–d). The tetraspanin family (CD9, CD63, and CD81) of EV markers, as well as the annexins (A5 and A7), are shown in Figure 4a. Notably, the LNCaP cells appear to express relatively higher levels of tetraspanins compared to the MSC (Figure 4a). Annexin A5 and A7 are cytosolic proteins [49] that are also differentially expressed but at mostly low levels, except for the enhanced expression of annexin A7 in the LNCaP cells. Interestingly, one of the most frequently reported EV-associated/marker proteins, flotillin-1 (Figure 4b), was expressed at predominantly low levels in both MSC and LNCaP EVs. The remaining cytosolic proteins commonly utilized and reported for EV identification include the programmed cell death 6 interacting protein (ALIX), as shown in Figure 4c, and the tumor susceptibility gene 101 protein (TSG101) and syntenin-1, highlighted in Figure 4d [50]. Both ALIX and TSG101 are soluble, cytosolic proteins that are associated with multivesicular bodies during EV biogenesis, while syntenin-1 is a peripheral inner membrane protein associated with cancer metastasis. All three cytosolic proteins showed variable expression levels between the two cell types. However, it was noted that the expression levels of these three proteins were markedly higher in the LNCaP samples compared to the MSC EVs. The presence of non-EV protein contamination in the samples is depicted by red arrows in Figure 4a,c. The MSC and the LNCaP EV samples showed the presence of serum albumin (higher in MSC) and endoplasmin (HSP90B1; higher in LNCaP). The MSC and the LNCaP EV samples were isolated and concentrated from donor cells using TFF. The non-EV protein contaminants may not have been efficiently removed during the TFF process, and/or the non-EV protein contaminants could be inherent to each EV sample’s protein corona [51]. Interestingly, other characteristic non-EV protein contaminants, such as cytochrome C1, major histocompatibility complex 1, calnexin, and apolipoprotein A-2, were not detected in either EV sample.

### 3.6. Small RNA (miRNA) Sequencing Analysis of LNCaP Cell EVs

Given that small RNAs are enriched in EVs [52], we profiled the small RNA cargo of LNCaP EVs at two facilities (A and B). LNCaP EVs were selected over MSC EVs for profiling due to their higher tetraspanin expression (Figure 4a) and budget constraints. Two QC methods were used at NIST post-small RNA isolation: Nanodrop and Agilent Bioanalyzer picochip, the latter recommended by the ISEV RNA position paper [53]. Nanodrop showed Vial #2 had ~2× higher yield than Vial #1, but Vial #1 had greater purity (A260/A280 ≈2) and fewer contaminants (higher A260/A230). Bioanalyzer profiles were similar across vials (Appendix A). While both facilities advertised sufficient read depth (>20 million reads per sample), Facility B samples contained more adapter reads (Appendix A)—a positive quality indicator—and showed the highest miRNA abundance (Appendix A), particularly in high-content samples. The absence of a small, degraded RNA peak in Facility B’s low-content samples suggests cleaner data (Appendix A). Differences in RNA type distributions and specific size patterns (e.g., fewer 33-nts peaks in Facility B high-content samples) highlight differences in library preparation methods.

Treemaps of differential miRNA expression (Figure 5a) reveal distinct expression patterns between facilities. In low-content samples, Facility A showed high expression of *hsa-let-7b-5p, hsa-let-7c-5p, hsa-miR-125b-5p, and hsa-let-7a-5p*, while Facility B enriched for *hsa-miR-99a-5p*, *hsa-let-7c-5p*, *hsa-miR-148a-3p*, and *hsa-let-7b-5p*. *hsa-let-7b-5p* was highly expressed in both (Figure 5a). In high-content samples, Facility A had high levels of *hsa-let-7b-5p*, *hsa-let-7c-5p*, and *hsa-miR-125b-5p*, while Facility B showed elevated *hsa-miR-99a-5p*, *hsa-miR-148a-3p*, and *hsa-let-7c-5p*, with *hsa-let-7c-5p* common to both (Appendix A). Prior studies have demonstrated that *hsa-let-7b-5p* is enriched in EVs and plays significant roles in intercellular communication [54,55]. Pearson correlation analysis (Figure 5b) shows that samples cluster by facility, with low- and high-content samples grouping within each facility. This suggests that while technical differences exist between facilities, sample preparation within each facility is reproducible. Facility A identified 694 miRNAs (Appendix A) and Facility B identified 740 miRNAs (Appendix A), with 485 miRNAs common to both (Figure 5c, Appendix A). Combining data from both facilities revealed 949 unique miRNAs (Appendix A). All identified miRNAs overlapped with the EVAtlas dataset ([36]; Appendix A). A complete comparison of sequencing results, expression data, and R code is available in the Appendix A.

## 4. Discussion and Conclusions

Orthogonal analytical methods were used to characterize the PSDs and PNCs of three commercially sourced materials: two representative test samples (LNCaP and MSC EVs) and one commercial exosome RM (GFP VLPs), for their suitability as RMs. The primary goal of this study was to determine whether these materials demonstrate measurement consistency sufficient for use as candidate EV reference materials. In addition, the protein and nucleic acid cargo, including the miRNA content specific to LNCaP EVs, were evaluated to assess how cargo variability affects RM suitability. Cryo-EM was used to assess morphology, and instrument reproducibility was assessed by PSD and PNC measurements across four different analytical methods: Cryo-EM, MRPS, PTA, and AF^4^. GFP VLP RM served as an experimental benchmark to assess instrument variability and potential measurement bias. MRPS yielded a larger mean particle diameter than Cryo-EM, and PTA yielded a broader PSD with a larger mean particle diameter. AF^4^ generated inconsistent PSD results across all EVs, including the GFP VLPs, with high replicate-to-replicate relative uncertainty likely due to sensitivity to particle volume, refractive index (RI), and dilution effects, which distinguish it from the other methods [17,29,41,44]. The mean PSD highlights the impact of the instrumental techniques or cell lines rather than equivalence among replicates.

Cryo-EM and MRPS demonstrate low between-sample variability and consistent, asymmetric particle size distributions, whereas PTA captured a broader range of particles but was sensitive to data-processing decisions (e.g., artifacts due to binning). AF^4^ showed substantial variability, particularly for MSC and GFP VLPs, suggesting its sensitivity to detecting EV subpopulations. PSD analysis also revealed that cancer cell-derived EVs exhibited lower uncertainty and higher reproducibility compared to stem cell-derived EVs. This observation must be validated in other tumors and stem cell-derived EVs. PNC measurements varied by up to two orders of magnitude across different cell lines and measurement methods, with AF^4^ showing the highest variability. These findings show that current commercial materials, including the GFP VLP RM, do not yet provide uncertainty-quantified benchmarks for PSD or PNC, underscoring the need for more robust EV RMs.

Overall, our findings clarify the capabilities and limitations of each method, defining which EV attributes can be measured reproducibly. Cryo-EM remains the gold standard for sizing, but its throughput is limited. Cryo-EM and MRPS provided the most reproducible PSD measurements under identical conditions. AF^4^ may be useful for detecting subpopulations; however, its utility requires further validation due to its variability. PTA remains valuable for broad particle profiling but benefits from standardized data-processing parameters. Because each platform relies on different particle properties, integrating data from multiple complementary methods is necessary to mitigate method-specific biases and obtain a reliable representation of size and concentration. Method harmonization or cross-platform validation is therefore essential when interpreting differences between PSD and PNC across datasets.

Each method relies on different physical attributes of the particle for measurement. For example, PTA and AF^4^ rely on light scattering for particle measurement, but the sample handling and analytical approaches differ. Cryo-EM and MRPS measure distinct particle properties; yet, their particle size distributions exhibit similar overall trends, with MRPS measurements showing a modest shift toward larger particle sizes, indicating systematic differences between the methods. Across all platforms, the samples displayed broad and heterogeneous size distributions (approximately 30 nm to 350 nm). The particle concentrations generally decreased as size increased, except for AF^4^ PSD. These observations reinforce the notion that no single method can fully characterize EVs with sufficient accuracy for RM development. Integrating data from multiple complementary techniques enables a more comprehensive characterization of candidate EV reference materials, enhancing their suitability for standardization and measurement validation. Sample-volume considerations are also important: MRPS requires <10 µL, while AF^4^ typically requires >100 µL; larger-volume methods may better capture heterogeneity, whereas smaller-volume methods preserve limited samples. Other platforms not evaluated here (e.g., super-resolution imaging, flow cytometry) may provide higher specificity and precision than PTA, MRPS, or AF^4^.

EV cargo characterization is a critical quality attribute for discriminating between EVs and non-EV particles and for RM development. The present study revealed that the types of EV-producing cells and sample purity strongly influence the detection and relative quantification of putative EV marker proteins. For example, we found striking differences in total protein between MSC and LNCaP during the preanalytical phase of sample preparation, which could have led to the enhancement or diminution of the detected EV-specific markers. Systematic purity checks are recommended to reduce downstream uncertainty and inform the suitability of producer cell types for RM generation. Additionally, the choice of EV isolation/purification method must be carefully selected, as it can significantly influence the proteomic profile.

Circulating EV miRNAs may reflect ongoing pathophysiological processes and predict patient outcomes [56]. RNA-seq captures a diverse range of small RNAs and miRNAs in EVs. Variability in EV miRNA detection across facilities also highlights the need for workflow standardization. Differences in abundance of shared miRNA across facilities (e.g., *hsa-let-7b-5p* and *hsa-let-7c-5p*) underline the need for controls to detect technical variation. Based on our findings, high-input samples, standardized library preparation, early monitoring for contaminants and degradation, and facility-specific normalization are recommended. In addition, spike-in controls like *U6* and *miR-16-5p* [57,58], or non-human RNA/ miRNA as QC controls [59] for library preparation and subsequent RNA sequencing studies may help; however, their expression profiles across diverse biological contexts need to be comprehensively characterized.

This study highlights some of the key challenges in developing EV RMs. Overall, an appropriate RM for EVs may consist of a suite of materials that capture EV attributes, such as PSD, PNC, protein content, and/or miRNA sequences, rather than a single universal material. Given the difficulties in establishing RMs and inconsistencies between measurement methods, extensive interlaboratory studies are crucial to achieving consensus on EV PSD and PNC. Leveraging EV preparation from commercial and non-profit labs accelerates RM development by utilizing their expertise and production capacity. In collaboration with other organizations, NIST will continue characterizing EV concentration, size distribution, and cargo content to develop uncertainty-quantified, fit-for-purpose RMs that support measurement standardization across the EV field.

## Figures and Tables

**Figure 1 biomolecules-16-00066-f001:**
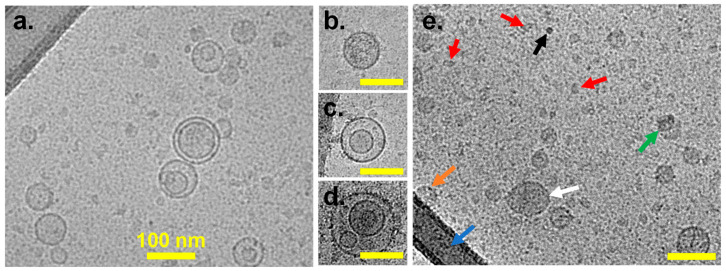
Representative Cryo-EM images of EVs: (**a**) Vesicles derived from MSCs; (**b**–**d**) examples from LNCaP of single vesicles (**b**), double vesicles (**c**), and multilayer vesicles (**d**); (**e**) LNCaP-derived EVs with single vesicles (white arrow), overlapping vesicles (green arrow), vesicle (orange arrow) overlapping thick carbon support (blue arrow), high-contrast contamination (black arrow), and small fragments < 30 nm (red arrow). Scale bars are 100 nm for panels (**b**–**d**).

**Figure 2 biomolecules-16-00066-f002:**
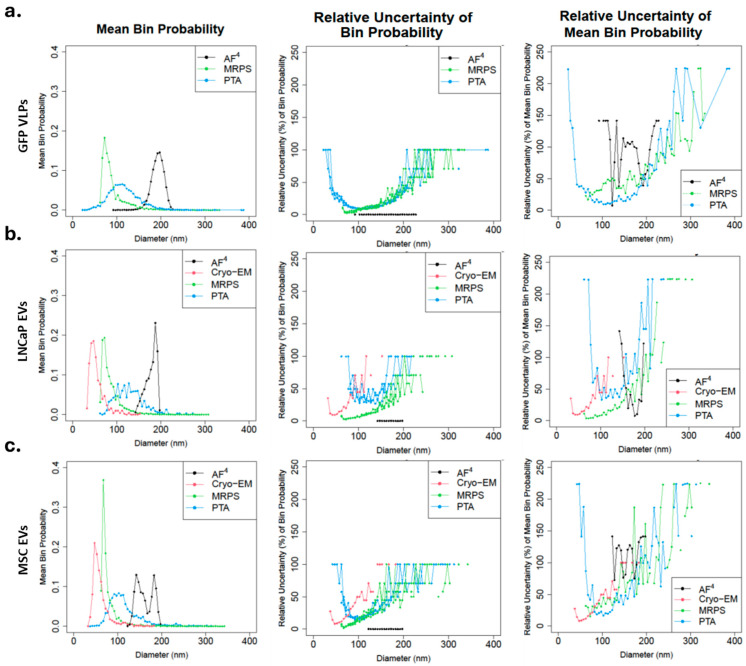
Mean PSD and their uncertainties by cell line. EV PSD was measured in triplicate for each method (AF^4^, Cryo-EM, MRPS, PTA) and then combined into a mean distribution by averaging the three probabilities of each replicate for three cell lines: ((**a**), **left**) GFP, ((**b**), **left**) LNCaP, and ((**c**), **left**) MSC. Mean PSD for AF^4^ (black), Cryo-EM (red), MRPS (green), and PTA (blue). Single measurement relative uncertainties for each independent replicate in % at each diameter (nm) for three cell lines: ((**a**), **center**) GFP, ((**b**), **center**) LNCaP, and ((**c**), **center**) MSC for AF^4^ (black), Cryo-EM (red), MRPS (green), and PTA (blue). AF^4^ single measurement relative uncertainty values are near zero. Replicate-to-replicate relative uncertainties in % at each diameter (nm) for the mean distributions of three cell lines: ((**a**), **right**) GFP, ((**b**), **right**) LNCaP, and ((**c**), **right**) MSC for AF^4^ (black), Cryo-EM (red), MRPS (green), and PTA (blue). Cryo-EM was not performed on GFP VLPs. Cryo-EM was not performed in triplicate; a single analysis of ≈500 particles was evaluated.

**Figure 3 biomolecules-16-00066-f003:**
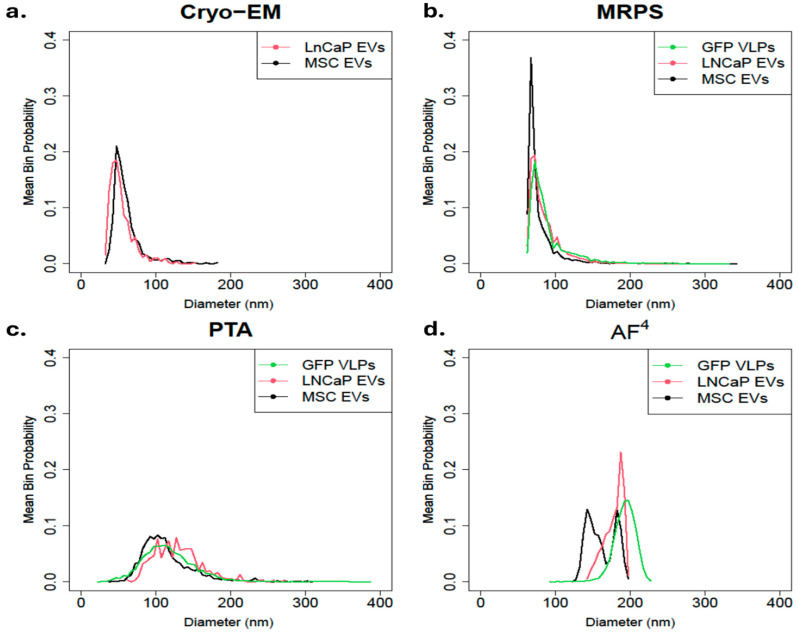
Comparison of mean PSD by method. Particle PSD was measured in triplicate for each cell line (LNCaP EVs, MSC EVs, and GFP VLPs) and then combined into a mean distribution by averaging the three probabilities of each replicate using four methods: (**a**) Cryo-EM, (**b**) MRPS, (**c**) PTA, and (**d**) AF^4^. Cryo-Em was not performed on GFP VLP. Cryo-EM was not performed in triplicate; a single analysis of ≈500 particles was evaluated.

**Figure 4 biomolecules-16-00066-f004:**
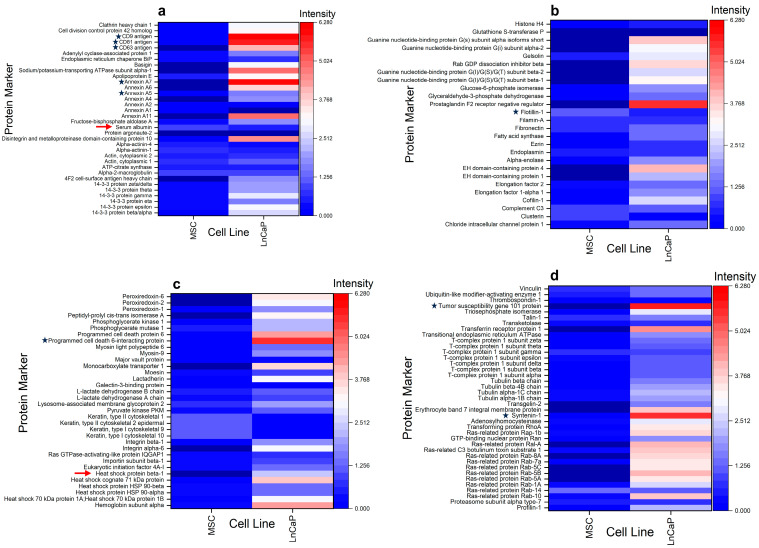
Heatmaps of 131 EV marker proteins differentially expressed and simultaneously identified in EVs from MSC and LNCaP cell lines by mass spectrometry-based proteomics. Data (n = 3 biological replicates for each cell type) were collected and analyzed using global proteome DIA ESI-LC-MS/MS conditions as described in the Experimental section. Heatmap rows show EV marker proteins identified in at least one of the cell types. Heatmap columns show normalized MS peak intensities for all cell types, scaled as shown. Lower and higher values than the normalized mean intensities (white) are represented in blue and red scales, respectively. (**a**–**d**) each show a subset of EV marker proteins with frequently identified small EV markers with a black star. Non-EV protein contaminations are depicted by red arrows (**a**,**c**). Heatmaps were generated via OriginPro 2023.

**Figure 5 biomolecules-16-00066-f005:**
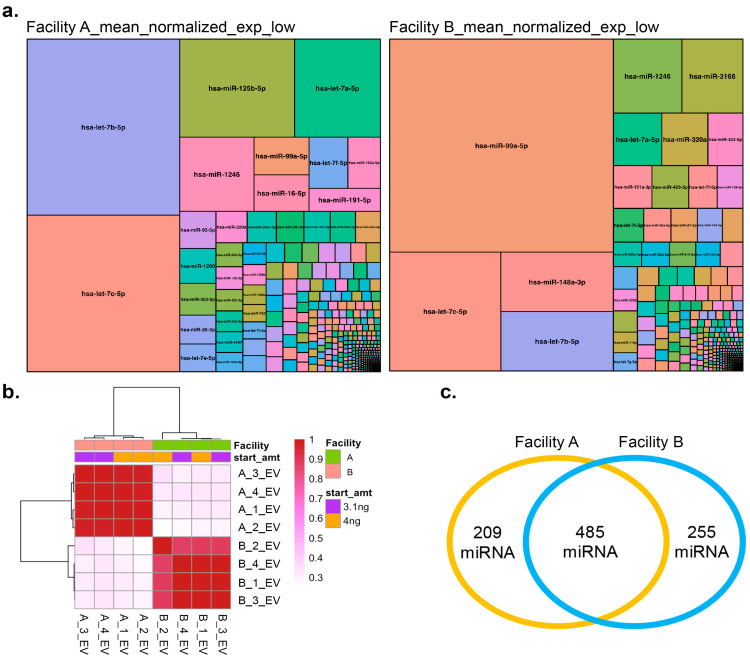
Analysis of miRNA seq results of small RNA profiles from two facilities (**a**) Differential expression of miRNA identified from the two facilities in the analysis presented as treemaps for low content (3.1 ng). Note: A complete List of each miRNA with differential expression profiles in a tabular format can also be found in the Appendix A. (**b**) Pearson correlation between the normalized counts of the samples. (**c**) Venn diagram showing the total and common miRNA between the two facilities.

**Table 1 biomolecules-16-00066-t001:** Particle number concentration (1/mL) of the control GFP VLPs (nominal value 3 × 10^9^/mL) and both cell line-derived EVs (MSC and LNCaP) as measured by three orthogonal techniques (PTA, AF^4^, and MRPS). Particle number concentrations are mean +/− 1 standard deviation: n = 3.

Technique\/Cell Line		Measurement Cutoff (nm)	GFP VLPs	MSC EVs	LNCaP EVs
Particle number concentration (1/mL)	PTA ^(a)^	25	(9 ± 1) × 10^9^	(20 ± 6) × 10^12^	(3.4 ± 0.6) × 10^12^
	AF^4^	30	(5 ± 1) × 10^10^	(3 ± 2) × 10^11^	(4.9 ± 0.3) × 10^10^
	MRPS	65	(8 ± 4) × 10^9^	(1.1 ± 0.4) × 10^12^	(7.1 ± 0.1) × 10^11^

^(a)^ Measurement cutoff for PTA is dependent upon sample and instrument settings. Please refer to the Section 2 for PTA setup.

## Data Availability

The original contributions presented in this study are included in the article/Appendix A. Further inquiries can be directed to the corresponding author.

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
