# Peer review of "Intermethod Characterization of Commercially Available Extracellular Vesicles as Reference Materials"

_biomolecules, 2025, doi:10.3390/biom16010066_

Round 1

Reviewer 1 Report

Comments and Suggestions for Authors

I enjoyed reviewing this article very much and recognise congratulate the authors for this elegant study. There is a very big need for reference materials and comparison of measurement devices in the extracellular vesicle research community. For this reason, this paper will be of broad research interest. My biggest concern is the inclusion of the commercially available standards which have raised concerns in the community for rather being viral like particles (VLP) than extracellular vesicles. Still, I believe this article is suitable for publication after addressing my comments below. I especially note the importance of removing data from the commercially available standard or at the very minimum revising most of the text to emphasize their VLP nature with more critical discussion about their use.

Major comments

  1. There are major concerns with use of the commercially available GFP “EV” standards from Millipore Sigma (SAE0193, Lot #000023251). According to the reviewer knowledge of how the standard was developed as well as the manufacturers product sheet (SAE0193pis Rev 02/22 NA,LN,GCY) with the relevant citation (PMID: 31337761) we know that eGFP labelling is produced by a gag-eGFP fusion protein. This citation is also provided by the authors in this paper as [36]. The issue is that this is widely recognised to be a very poor standard because gag-eGFP makes these viral-like particles and not EVs as gag will never be a natural or physiological cargo of human EVs. For this reason, data generated with this standard should be removed from the study or at the very minimum all results should be reframed as obtained using gag-eGFP VLP standards so that the authors and field can decide for themselves whether they believe a VLP standard may be a suitable reference for EV studies.
  2. Please adjust figure 3 y axis in all subpanels to a maximum of 0.4 so that the panel B doesn’t cut-off the MSC line.

Minor comments:

  1. On lines 39-41 it is claimed that “The three main subtypes of EVs are ectosomes, exosomes, and apoptotic bodies [5], which are differentiated based on their biogenesis, release pathways, size, content, and function”. However, the field of EV research strongly discourages the classification of EVs based on biological process such as apoptotic bodies, migrasomes, large oncosomes, etc… and instead encourages biogenesis classification into ectosomes (plasma membrane derived) and/or exosomes (endosomal compartment, MVB/ILV derived) when it is possible to make such distinction. This recommendation is outlined in the 2023 MISEV of which one of the present paper authors is also a co-author (PMID: 38326288). With this in mind I kindly ask, the authors to please change the original text to instead indicate two main subtypes of EVs are ectosomes and exosomes differentiated based on their biogenesis, release pathways, size and content [PMID: 38326288].
  2. On lines 120-122 the use of 1% Tween 20 is justified due to two studies showing that it doesn’t alter EVs. Actually besides the two studies cited another earlier study (PMID: 26264754) should also be cited having shown this finding first and to further justify that 1% Tween 20 doesn’t disrupt EVs.
  3. In lines 159, 249, 268 it appears that these are the titles of methods subsections but incorrectly labeled. If this is so please correct by marking them as subsection titles accompanied by their subsection numbers (eg 4. Particle Asymmetric Flow Field Flow Fractionation-Multi-Angle Light Scattering).
  4. In lines 513 and 514 particles per protein are reported. However the units should be checked, currently it is listed as for example “2.1 x 109 1/µg protein” but instead it should be “2.1 x 109/µg protein”. The number 1 is not needed before /ug.
  5. Figure 4 is extremely blurry/low resolution making it hard to read the proteins identified or find the arrows showing the contaminating proteins. Perhaps this is due to the reviewer version available to me but the publication should have a higher resolution.

Author Response

Thank you for you valuable comments to improve our manuscript. The responses to your comments are populated below. 

Major comments

  1. There are major concerns with use of the commercially available GFP “EV” standards from Millipore Sigma (SAE0193, Lot #000023251). According to the reviewer knowledge of how the standard was developed as well as the manufacturers product sheet (SAE0193pis Rev 02/22 NA,LN,GCY) with the relevant citation (PMID: 31337761) we know that eGFP labelling is produced by a gag-eGFP fusion protein. This citation is also provided by the authors in this paper as [36]. The issue is that this is widely recognised to be a very poor standard because gag-eGFP makes these viral-like particles and not EVs as gag will never be a natural or physiological cargo of human EVs. For this reason, data generated with this standard should be removed from the study or at the very minimum all results should be reframed as obtained using gag-eGFP VLP standards so that the authors and field can decide for themselves whether they believe a VLP standard may be a suitable reference for EV studies.

We thank the reviewer for highlighting the important distinction that the eGFP-labeled material corresponds to gag-eGFP viral-like particles (VLPs) rather than physiological EVs. We agree that gag is not a natural EV cargo and that this limits the suitability of these particles as a biological EV reference. However, we respectfully request that to retain the GFP EV data for the following reasons:

  • The product has been cited in >70 peer-reviewed publications, including EV method-development and technology-comparison studies. Its extensive use reflects its practical role as a consistency standard nanoparticle assay benchmarking, even though it is not a physiological EV surrogate.
  • Our use of the gag-eGFP standard in this study is restricted to technical verification of detection sensitivity, assay linearity, and instrument comparability, not as a model for EV biology. These specific analytical purposes are precisely where VLP-type standards remain informative and accepted in the field.
  • The conclusions of the study are based on multiple orthogonal EV-relevant samples. The gag-eGFP VLP data are not used to draw biological conclusions, but rather to support the characterization of assay performance. Removing these results would eliminate a useful benchmarking component without affecting the biological findings.

We have explicitly added the following disclaimer per reviewer’s recommendation “The GFP labeling in these GFP EVs is generated through a viral gag-eGFP fusion, meaning the material behaves like viral-like particles (VLP) rather than physiological EVs because gag is not a natural EV cargo. For this reason, we used the material solely as a technical reference to assess detection sensitivity, assay linearity, instrument comparability, and nanoparticle assay benchmarking. Thus, GFP EVs are used to support assay performance, rather than to draw biological conclusions.” Lines 82 -88 We have reiterated the notion again. We have therefore added explicit clarification in the revised text (lines 328 to 330) that this material represents a gag-eGFP EV, not a native EV control, and we now interpret the associated results accordingly.

  1. Please adjust figure 3 y axis in all subpanels to a maximum of 0.4 so that the panel B doesn’t cut-off the MSC line.

We have adjusted Figure 3 y-axis in all subpanels to a maximum set at 0.4 as per the reviewer recommendation. Line 477

 Minor comments:

  1. On lines 39-41 it is claimed that “The three main subtypes of EVs are ectosomes, exosomes, and apoptotic bodies [5], which are differentiated based on their biogenesis, release pathways, size, content, and function”. However, the field of EV research strongly discourages the classification of EVs based on biological process such as apoptotic bodies, migrasomes, large oncosomes, etc… and instead encourages biogenesis classification into ectosomes (plasma membrane derived) and/or exosomes (endosomal compartment, MVB/ILV derived) when it is possible to make such distinction. This recommendation is outlined in the 2023 MISEV of which one of the present paper authors is also a co-author (PMID: 38326288). With this in mind I kindly ask, the authors to please change the original text to instead indicate two main subtypes of EVs are ectosomes and exosomes differentiated based on their biogenesis, release pathways, size and content [PMID: 38326288].

We agree with the recommendation and have revised the text accordingly to reflect the two main EV subtypes, ectosomes and exosomes, consistent with the 2023 MISEV guidelines, and we have removed the previous reference to apoptotic bodies as a primary subtype and updated the text (Lines 39 to 41) along with the reference.

  1. On lines 120-122 the use of 1% Tween 20 is justified due to two studies showing that it doesn’t alter EVs. Actually besides the two studies cited another earlier study (PMID: 26264754) should also be cited having shown this finding first and to further justify that 1% Tween 20 doesn’t disrupt EVs.

We agree and have updated the reference by adding the recommended reference. (Line 130)

  1. In lines 159, 249, 268 it appears that these are the titles of methods subsections but incorrectly labeled. If this is so please correct by marking them as subsection titles accompanied by their subsection numbers (eg 4. Particle Asymmetric Flow Field Flow Fractionation-Multi-Angle Light Scattering).

We have updated all these as subsections; changes are reflected in lines 169, 251, 273.

  1. In lines 513 and 514 particles per protein are reported. However the units should be checked, currently it is listed as for example “2.1 x 109 1/µg protein” but instead it should be “2.1 x 109/µg protein”. The number 1 is not needed before /ug.

We agree and we have updated these changes (lines 516 and 517)

  1. Figure 4 is extremely blurry/low resolution making it hard to read the proteins identified or find the arrows showing the contaminating proteins. Perhaps this is due to the reviewer version available to me but the publication should have a higher resolution.

We have updated the figure with an hq tiff image (line 531); to enhance the figure we have also removed the red box and replaced with black stars (line 531). Additionally, we have updated the figure legend to reflect this change. We can also forward the hq tiff image to the editors if it is a concern.

Reviewer 2 Report

Comments and Suggestions for Authors

Review on manuscript

Intermeasurement Characterization of Commercially Available Extracellular Vesicles as Reference Materials

The work presented is truly excellent, interesting, and addresses a very important and topical issue. One of the bottlenecks in EV research is indeed standardization and the production of appropriately controlled characterization methods and reference materials.

I find the comparison of the four methods for determining particle size distribution and particle number concentration very interesting. However, this type of comparison is not new. For example, Witwer et al, compared four orthogonal technologies for sizing, counting, and phenotyping extracellular vesicles (EVs) and synthetic particles: single‐particle interferometric reflectance imaging sensing (SP‐IRIS) with fluorescence, nanoparticle tracking analysis (NTA) with fluorescence, microfluidic resistive pulse sensing (MRPS), and nanoflow cytometry measurement (NFCM). Nieuwland et al investigated size distribution of erythrocyte-derived EVs using nanoparticle tracking analysis, resistive pulse sensing, and electron microscopy, small-angle X-ray scattering (SAXS) and size exclusion chromatography coupled with dynamic light scattering detection.

I have some critical question and comments:

  1. I do not understand Figure 1 (a) of MSC-EV. Several vesicles should be shown.
  2. Based on the results, it seems that variability due to the applied technique (Cryo-EM, MRPS, PTA, AF4) is the determinant in particle size distribution and not the type of EV (MSC-EV, LNCaP-EV or GFP-EV). Is that possible?
  3. Table1: Particle number concentration of the three types of EVs measured by PTA, AF4 and MRPS. As I see, PTA and MRPS approaches the control value (GFP-EV with nominal value 3 × 109 1/mL). For LNCaP-EV there was a difference of two orders of magnitude in function of method. I don’t guess LoD in this case is correct, is rather the cut-off limitation of the method. LoD should be related to the lowest concentration which is still detected by the given method.
  4. The discussion should be more precise. I don't feel there is a specific message or statement regarding the purpose and outcome of the work.

Despite all this, the work is interesting and merit for publication after corrections.     

Author Response

Thank you for your valuable input to imporve our manuscript and for this observation. We agree that method comparison studies in the EV field have important precedents, including the work by Witwer et al. and Nieuwland et al. Our study builds on these foundations by evaluating a different set of commercially sourced materials and focusing on practical implications for assay performance and inter-method variability.

  1. I do not understand Figure 1 (a) of MSC-EV. Several vesicles should be shown.

We have updated Figure 1 to clarify our intent which was to show various types/ subpopulations of EVs from Cryo-EM analysis. We have included various images of both MSC and LNCaP EV along with various subpopulation of EVs, line numbers  330. In addition, we have updated the text to reflect the changes (lines 319-322) and figure legend (331-333)

  1. Based on the results, it seems that variability due to the applied technique (Cryo-EM, MRPS, PTA, AF4) is the determinant in particle size distribution and not the type of EV (MSC-EV, LNCaP-EV or GFP-EV). Is that possible?

While the results may suggest that technique-dependent variability (Cryo-EM, MRPS, PTA, AF4) is a major driver of differences in the measured particle size distributions, it is unlikely to be the sole determinant of this phenomenon. In practice, both the analytical method and the material itself contribute to variability as we have shown in this paper at different instances. Each technique has distinct operating principles, sensitivities, and biases that can shift the apparent size distribution which we have pointed out at various segments of the paper. At the same time, the EV preparations (MSC-EV, LNCaP-EV, and GFP-EV) differ in their biogenesis pathways, cargo composition, and physical properties, which can influence their behavior. We have shown in this paper with various techniques that MSC EV show higher variability in sizing and counts. Thus, the observed variability likely reflects an interplay between method-specific effects and intrinsic differences among the EV types, rather than being driven exclusively by either factor.

  1. Table1: Particle number concentration of the three types of EVs measured by PTA, AF4and MRPS. As I see, PTA and MRPS approaches the control value (GFP-EV with nominal value 3 × 109 1/mL). For LNCaP-EV there was a difference of two orders of magnitude in function of method. I don’t guess LoD in this case is correct, is rather the cut-off limitation of the method. LoD should be related to the lowest concentration which is still detected by the given method.

We appreciate this comment by the reviewer and agree that measurement cutoff is a more appropriate term than LoD for describing the intent of this experiment. We did not explicitly determine a limit of detection (e.g., based on three standard deviations above background); rather, our goal was to illustrate the effective lower measurement boundary imposed by each technique. Therefore, “measurement cutoff” more accurately reflects the methodological limitation being assessed. We have updated the table that reflects the recommendation (Lines 508- 510) and pertinent  text where LoD is mentioned (including the Supplemental Section)

  1. The discussion should be more precise. I don't feel there is a specific message or statement regarding the purpose and outcome of the work.

Thank you for this helpful comment. We have revised the Discussion section with paragraph-by-paragraph edits to improve clarity and precision. Redundant statements were removed, and we added explicit statements that communicate the specific purpose, key findings, and recommendations emerging from the work. We believe these changes strengthen the overall message and provide a more focused and coherent Discussion (lines 609 – 696).

Round 2

Reviewer 1 Report

Comments and Suggestions for Authors

I thank the authors for addressing most of the comments. I believe my comments 2-7 have all been fully addressed.

Regarding comment 1 there is still fundamental issue with keeping the term EV for the gag-eGFP VLP data and as such I can’t support acceptance of the manuscript without further revision. While I concede with the authors about leaving the data in the manuscript agreeing that gag-eGFP VLPs can still serve as reference materials I do outline below the 2 changes that must be addressed:

  1. Even with the disclaimer and clarification indicated by the authors, the term GFP EV is kept throughout the text (>20 times). Conversely, GAG or VLP only appear briefly in the disclaimer. Continuing using GFP EV nomenclature invalidates the disclaimer because the particles are repeatedly termed EVs. Instead, the full manuscript should consistently label them as “gag-eGFP VLPs”, borrowing the term used by the authors in their response 2nd bullet point, or I can suggest “eGFP VLPs”, “GFP VLPs”.
  2. Figures should show “gag-eGFP VLPs”, “eGFP VLPs”, “GFP VLPs” for this data, currently “GFP” or “GFP EV” is used.

Author Response

We thank the reviewer for the additional feedback and for recognizing that comments 2–7 have been fully addressed. We also appreciate the reviewer’s continued engagement with comment 1. In response to the remaining concern regarding the nomenclature of the gag-eGFP VLP data, we have implemented all requested revisions. 

1. Even with the disclaimer and clarification indicated by the authors, the term GFP EV is kept throughout the text (>20 times). Conversely, GAG or VLP only appear briefly in the disclaimer. Continuing using GFP EV nomenclature invalidates the disclaimer because the particles are repeatedly termed EVs. Instead, the full manuscript should consistently label them as “gag-eGFP VLPs”, borrowing the term used by the authors in their response 2nd bullet point, or I can suggest “eGFP VLPs”, “GFP VLPs”.

We have replaced all instances of “GFP EV” and related variations with the terminology recommended by the reviewer, GFP VLPs. The manuscript now consistently refers to these particles as GFP VLPs where appropriate. We also made appropriate modifications to the language in certain cases to reflect the change. 

Removed EVs from the following lines and replace with VLPs.

Lines 82, 111, 154, 327, 344, 346, 363, 367, 392, 471, 472, 476, 481, 483, 486, 490, 494, 508, 614, 620, 624, 632, 638, 846, 853

Made the following changes to accommodate the VLP language:

Line 87 Added "and henceforth referred to as GFP VLPs"

Line 88 Removed EVs, replaced with "materials"

Line 108 Removed GFP EVs, replaced with "green fluorescent protein exosome" 

2. Figures should show “gag-eGFP VLPs”, “eGFP VLPs”, “GFP VLPs” for this data, currently “GFP” or “GFP EV” is used.

No figure now uses the terms “GFP” or “GFP EV” to describe these materials (Table 1, Figs. 2 and 3 reflect the change).

We appreciate the reviewer’s careful attention to this issue and the constructive guidance provided.

Round 3

Reviewer 1 Report

Comments and Suggestions for Authors

Thank you for addressing all my comments. In the current form I believe this wonderful study should be accepted and will be of great use and interest for EV researchers.